# Theoretical Study on the Copper-Catalyzed *ortho*-Selective C-H Functionalization of Naphthols with *α*-Phenyl-*α*-Diazoesters

**DOI:** 10.3390/molecules28041767

**Published:** 2023-02-13

**Authors:** Xiaoli Zhu, Xunshen Liu, Fei Xia, Lu Liu

**Affiliations:** 1School of Chemistry and Molecular Engineering, East China Normal University, 500 Dongchuan Road, Shanghai 200241, China; 2Shanghai Engineering Research Center of Molecular Therapeutics and New Drug Development, East China Normal University, Shanghai 200062, China; 3NYU-ECNU Center for Computational Chemistry at New York University, East China Normal University, 3663 Zhongshan Road, Shanghai 200062, China

**Keywords:** copper catalysis, *ortho*-C(sp^2^)-H bond functionalization, naphthols, diazo compounds, metal carbene, density functional theory (DFT) calculations

## Abstract

The aromatic C(sp^2^)-H functionalization of unprotected naphthols with *α*-phenyl-*α*-diazoesters under mild conditions catalyzed by CuCl and CuCl_2_ exhibits high efficiency and unique *ortho*-selectivity. In this study, the combination of density functional theory (DFT) calculations and experiments is employed to investigate the mechanism of C-H functionalization, which reveals the fundamental origin of the site-selectivity. It explains that CuCl-catalyzed *ortho*-selective C-H functionlization is due to the bimetallic carbene, which differs from the reaction catalyzed by CuCl_2_ via monometallic carbene. The results demonstrate the function of favourable H-bond interactions on the site- and chemo-selectivity of reaction through stabilizing the rate-determining transition states in proton (1,3)-migration.

## 1. Introduction

The metal-carbenes, normally generated from diazo compounds via the catalysis of transition-metals, have been widely used in organic synthesis as one of the most significant reactive intermediates due to the versatile transformations [1,2,3,4,5,6,7,8,9,10,11,12,13,14,15,16], such as cycloaddition, cyclopropanation, ylide formation and rearrangement, O-H bond insertion, N-H bond insertion, and C-H bond functionalization [17,18,19,20,21,22,23,24,25,26,27,28,29,30,31]. Thus, it is highly desirable to develop a new reactional methodology by using metal-carbenes as the key species in synthetic chemistry. On the other hand, because phenyl rings, such as benzene, phenol, aniline and their derivatives, occur widely in natural products, bioactive molecules and drugs are important platforms in organic synthesis. The highly site-selective aromatic C(sp^2^)-H bond functionalization of phenyl ring without the directing groups is still challenging, especially for the phenol derivatives [32,33,34,35,36,37,38]. In this field, due to its high activity in organic reactions, metal-carbene species have the advantage in the activation of inert C-H bond, while still meeting the chemo- and region-selectivities. For example, the phenolic O-H insertion occurred during the reactions of phenol derivatives with diazoesters in the presence of various metal catalysts such as Rh, Fe, Ru, Cu, and Pd (Figure 1a) [39,40,41,42,43,44]. In 2014, we disclosed the first gold-catalyzed aromatic C(sp^2^)-H bond functionalization of free phenols with diazo compounds in high efficiency with excellent *para*-selectivity [45]. Later, we reported the challenging *ortho*-selective aromatic C(sp^2^)-H bond alkylation of phenols with diazoesters by using B(C_6_F_5_)_3_ and naphthols with diazoesters by using gold catalyst (Figure 1b) [46,47]. Nemoto disclosed the similar gold catalyzed C(sp^2^)-H bond functionalization/cyclization of *β*-naphthols with diazoesters [48]. However, the catalysts used in these reactions, like gold complexes and B(C_6_F_5_)_3_, are expensive. Thus, the development of inexpensive and abundant catalysts to replace the noble catalysts is a long-term need. Furthermore, understanding the origins of new catalytic systems in carbene chemistry is also very important, which could guide the design of new reactions and new catalysts.

Compared to other commonly used transition-metals in organic synthesis, copper represents a type of ideal catalyst for chemical reactions. It attracts much attention due to its low-toxicity, low cost, ready availability, and its benign environmental impact [49,50,51,52,53]. It has been used in carbene transfer reactions for half a century. However, the O-H insertion was the major reaction when phenol derivatives reacted with diazo compounds in the presence of copper [54,55,56,57]. In 1952, Yates reported the reaction of phenols and diazo compounds under copper catalyst, delivering O-H insertion product as the primary one along with the side product via the *ortho*-selective C-H bond functionalization/cycliztion [54]. Recently, Zhou disclosed an elegant asymmetric copper-catalyzed O-H insertion of phenols [56]. Apart from experimental investigations, mechanistic investigation of the O-H insertion by copper-catalyzed carbenes transferred from diazo compounds were also performed [58,59,60,61,62]. Pérez et al. examined the mechanism of copper-catalyzed O-H insertion reaction, in which they investigated the ligand effects of Tp^X^ (hydrotris(3,5-dimethylpyrazolyl)borate and its derivatives) on chemoselectivity by experiments [58]. Yu et al. explored the detailed mechanism of O-H insertion of diazoacetates under copper (I) by DFT calculation [59]. They discovered that the (1,2)-H migration favored the copper-associated ylide pathway, in which the water molecule acted as an effective proton shuttle for the (1,2)-H shift in Cu-catalyzed O-H insertion. In contrast to the above cases, we recently reported a copper-catalyzed *ortho*-selective C-H bond alkylation of naphthols and phenols that provided an important route for the synthesis of *ortho*-substituted phenol derivatives [63]. However, the detailed mechanism of copper-catalyzed C(sp^2^)-H bond functionalization is still unknown due to the more complex structures and versatile valence states of copper (Figure 1c). In our previous work, we studied the competitive pathways of gold catalyzed C-H bond functionalization and O-H bond insertions of phenols with diazo compounds via the combination of DFT calculations and experiments [61,62]. In this report, both CuCl and CuCl_2_ were efficient catalysts for this reaction, providing a good yield of the *ortho*-selective products with excellent site-selectivity (Table 1). In this study, we investigated the mechanism of site-selective C(sp^2^)-H bond functionalization of 1-naphthol and *α*-diazoacetate catalyzed by CuCl and CuCl_2_ by combining DFT calculations and experiments.

Previously, we reported a mechanistic study on how to yield the active Cu-carbenes from diazo compounds with CuCl and CuCl_2_ [64]. The DFT calculations revealed that the most stable structure of CuCl in the solution was dimer, while that of CuCl_2_ was monomer. Then, the decomposition of *α*-phenyl-*α*-diazoester under CuCl generated the bimetallic carbene, while the monometallic Cu(II)-carbene was obtained in the presence of CuCl_2_ (Figure 1). Therefore, the two types of Cu-carbenes will be used as the precursors for the site-selective C(sp_2_)-H bond functionalization of naphthols.

## 2. Results and Discussion

Under the catalysis of transition metal Cu, we studied the mechanisms of C-H bond functionalization between diazo compounds and naphthols to generate the *ortho*-substituted products. Previous experimental studies reported that the transition metal complexes initially reacted with the methyl phenyl diazoacetates to form metal carbenes by releasing the N_2_ molecules [65,66,67]. The metal carbenes and naphthols were regarded as the precursors for the C-H insertion. Thus, the Cu-carbenes and naphthols are used as the reaction precursors in our DFT calculations. The relative energies of all intermediates and transition states to the sum of precursors are calculated and presented in all figures. The red and pink lines denote the lowest energy pathways of *ortho*-C-H and *para*-C-H insertions, respectively.

### 2.1. The C-H Insertion of Naphthols Catalyzed by the (CuCl)_2_ Dimer

As discussed, the DFT calculations revealed that CuCl preferred the dimer to the monomer in the solution [64]. Assuming the (CuCl)_2_ dimer as an effective catalyst, it reacts with the diazo compound to yield the bimetallic Cu carbene. Figure 2 shows the calculated free energy profiles for the detailed reaction pathways of the C(sp^2^)-H bonds of naphthols inserted by bimetallic Cu carbenes. The bimetallic Cu carbenes and naphthols undergo the electrophilic addition to form the intermediate **1-Int-o2** through the ortho-substituted transition state **1-TS-o1** with a barrier of 11.9 kcal mol^−1^. Another reaction pathway of the addition at the para-C(sp^2^) of naphthols via **1-TS-p1** has a higher barrier of 12.7 kcal mol^−1^. In the two addition pathways, the protons of the aromatic C-H bonds transfer to the carboxyl oxygen atoms via the optimized five-membered ring transition states **1-TS-o3/p3**. In this case, the protons of **1-Int-o2/p2** at the ortho- and para-carbons of naphthols move to the carbonyl oxygens to form the enols **1-Int-o4/p4**, which overcome the activation barriers of 8.5 and 8.6 kcal mol^−1^, respectively.

There are two possible reaction pathways from **1-Int-o4** to yield the final product **Pro-o8**. One is the conversion of **1-Int-o4** into **1-Int-o5*** through the (CuCl)_2_ dimer dissociating into the solution by absorbing an energy of 7.0 kcal mol^−1^. The second pathway is that the (CuCl)_2_ dimer in **1-Int-o4** undergoes the (1,3)-migration to the phenyl group to yield the less stable enol **1-Int-o5**. However, the formation of **1-Int-o5** is more favorable than **1-Int-o5*** by 4.9 kcal mol^−1^ in energy. For the para-C-H insertion of naphthols, the reaction process is like the ortho-C-H insertion. The enol complex **1-Int-p4** could be further stabilized through the intramolecular (1,3)-migration of the (CuCl)_2_ dimer, leading to the enol intermediate **1-Int-p5** rather than to **1-Int-p5***. Subsequently, the metal catalysts participate in the two-water assisted (1,3)-H migration via the eight-membered ring transition states **1-TS-o6-2w** and **1-TS-p6-2w**. Comparison of the calculated energies of the optimized TSs for proton transfer suggests that the energy of **1-TS-o6-2w** is lower than that of **1-TS-p6-2w** by 1.4 kcal mol^−1^. In this case, the proton transfer from the hydroxyl group to the ortho-position carbon of naphthol through **1-TS-o6-2w** leads to the formation of **1-Int-o7** with a barrier of 18.7 kcal mol^−1^. Finally, the (CuCl)_2_ dimer of **1-Int-o7** dissociates into the solution to yield the final product **Pro-o8** with the calculated energy of −35.8 kcal mol^−1^.

### 2.2. The Key H-Bond Interactions Formed with (CuCl)_2_

To further explain the site-selectivity of the bimetallic Cu carbene catalyzed C(sp^2^)-H functionalization of naphthols with diazo compounds, we focus on the two important elementary steps during the activation of the ortho-C-H bond of naphthols by the bimetallic Cu carbenes: the electrophilic addition of bimetallic Cu carbenes and naphthols, and the two water-assisted (1,3)-proton transfer with the participation of Cu catalyst, which is the rate-determining step of the reaction. It is found that both the electrophilic addition and the (1,3)-H migration at the ortho-sites of naphthols are superior to that of the para-sites. The optimized structures and energies of **1-TS-1o**, **1-TS-1p**, **1-TS-o6-2w**, and **1-TS-6p-2w** are shown in Figure 3. It is noted that the electrophilic addition at the ortho-C(sp^2^) of naphthol has a lower barrier of 11.9 kcal mol^−1^ relative to that at the para-C(sp^2^), which is consistent with the known experiment results [63]. The **1-TS-o1** is stabilized by the O-H···Cl H-bond interaction between the hydroxyl group of naphthol and the Cl atom of bimetallic Cu carbene, with the H···Cl distance of 2.14 Å, as shown in Figure 3. 

Additionally, the H-bond interactions also play an important role in the two water-assisted proton transfer. The remote (1,3)-H migration needs two water molecules as a shuttle rather than one, which has been demonstrated in our previous study on the C-H insertion of phenols by Au-carbenes [61,62]. There are two kinds of H-bonds formed in the transition state **1-TS-o6-2w**, including the O-H···Cl H-bond interaction formed from the shuttle water and the Cl atom of the (CuCl)_2_ dimer, and the O-H···O H-bond interaction between the hydroxyl group of naphthol and the shuttle water. Due to the stabilization of the formed H-bonds in **1-TS-o6-2w**, the barrier of the (1,3)-proton shift is only 18.7 kcal mol^−1^, lower than that of 20.1 kcal mol^−1^ of **1-TS-p6-2w**, leading to the ortho-substituted products. Thus, the presence of H-bond interactions in the ortho-C-H functionalization reduces the energy barriers of the electrophilic addition of the metal carbenes with naphthols and the (1,3)-proton transfer process.

### 2.3. The C-H Insertion of Naphthols Catalyzed by the CuCl Monomer 

We also studied the C(sp^2^)-H bond insertion mechanism of naphthols catalyzed by the CuCl monomer instead of the (CuCl)_2_ dimer. Figure 4 displays the calculated free energy profiles of reaction pathways of the C-H bonds of naphthols inserted by the monometallic Cu carbenes. The first step remains the addition of the monometallic carbenes and naphthols and the calculated results indicate that the electrophilic addition at the para-C(sp^2^) of naphthols occurs via the transition state **1′-TS-p1** with a lower energy barrier of 15.6 kcal mol^−1^. This is lower than that of 17.6 kcal mol^−1^ at the ortho-C(sp^2^) via **1′-TS-o1**. Thus, the addition through **1′-TS-p1** is more kinetically favorable and not consistent with the experimental observations. If **1′-Int-o4** releases the moiety of the CuCl monomer into the solution, it yields a more stable intermediate **1′-Int-o5*** with the exothermicity of 2.8 kcal mol^−1^. However, the (1,3)-migration of CuCl to the phenyl group in **1′-Int-o4** leads to the enol **1′-Int-o5** with an exothermic energy of 5.9 kcal mol^−1^. Because **1′-Int-o5** is more stable than **1′-Int-o5*** by 3.1 kcal mol^−1^, the catalyst CuCl participates in the proton transfer. Also, **1′-Int-p4** transforms to the **1′-Int-p5** through the (1,3)-migration of the monomer CuCl instead of the dissociation of CuCl to form **1′-Int-p5***. Furthermore, the barrier from **1′-Int-o5** to **1′-TS-o6-2w** is high (up to 28.2 kcal mol^−1^)and considered as the rate-determining step, which is higher than that of **1′-Int-p5** to **1′-TS-p6-2w** by 5.8 kcal mol^−1^. Such a high barrier is counter to the experimental results. As such, the reaction pathways catalyzed by the monometallic carbenes of CuCl are excluded [63]. In summary, the calculated reaction pathways of C-H bond functionalization of naphthols catalyzed by the bimetallic and monometallic carbenes account for the site-selectivity of the C-H bond functionalization by the mild catalyst CuCl. The calculated results of these two steps show that the Cu-catalyzed C-H bond functionalization of naphthols with diazo esters is more inclined to obtain the ortho-substituted products.

### 2.4. The C-H Insertion of Naphthols Catalyzed by the CuCl_2_ Monomer

Figure 5 shows the calculated free energy profiles for the possible C-H bonds insertion pathways of naphthols catalyzed by the monometallic carbenes of CuCl_2_. The difference, when compared to Cu(ǀ), is the energy barriers of the electrophilic addition at the ortho-C(sp^2^) of naphthols, which is higher than that at the para-C(sp^2^) by 2.8 kcal mol^−1^. For ortho-C-H insertion, it is an endothermic process when the CuCl_2_ moiety of **2-Int-o4** migrates from the C=C double bond to the phenyl group to generate **2-Int-o5*** via (1,3)-migration of the monomer CuCl_2_. The **2-Int-o4** can transform to the key intermediate **2-Int-o5** through the CuCl_2_ monomer as it dissociates into the solution by releasing an energy of 4.6 kcal mol^−1^, which is relatively more feasible and stable compared to the formation of **2-Int-o5***. Like the process of ortho-C-H insertion, the enol complex **2-Int-p4** can be further stabilized by releasing CuCl_2_ into the solution, leading to a free enol **2-Int-p5** rather than **2-Int-p5***. Subsequently, the two water-assisted (1,3)-H migration without the participation of CuCl_2_ via the eight-membered ring transition states **2-TS-o6-2w** and **2-TS-p6-2w** are the pivotal steps in the ortho-C-H and para-C-H insertions, with the calculated barriers of 19.8 and 22.2 kcal mol^−1^, respectively. The activation energy barrier of **2-TS-p6-2w** is higher than that of **2-TS-o6-2w** by 2.4 kcal mol^−1^, implying that the C(sp^2^)-H insertion catalyzed by Cu(ǁ) prefers the ortho-selective product **Pro-o8** to the para-substituted product **Pro-p8**, which is in accordance with previous experimental results [58,59,60,61,62].

The structures of the key transition states **2-TS-1o** and **2-TS-1p** of the electrophilic additions in the two pathways are shown in Figure 6. The calculated distance between the C1 atom of the copper-carbene and the C2 atom of naphthol in **2-TS-1p** is longer than that of **2-TS-1o**. Although neither **2-TS-1o** nor **2-TS-1p** has H-bonds formed, the H-bond interaction of O-H···O between the hydroxyl group of naphthol and the water molecule in **2-TS-o6-2w** plays a crucial role in stabilizing the proton transfer through the Cu-free pathways, with a distance of 1.60 Å. The **2-TS-o6-2w** has a lower barrier than that of **2-TS-p6-2w**, which could be the key to control the chemo- and site-selectivity of the C-H bond insertion of naphthols catalyzed by Cu(ǁ).

### 2.5. Experimental Reactivity of 1-Methoxynaphthalene Catalyzed by Cu Catalysts

To further prove the Cu-catalyzed ortho-C-H insertion mechanism of naphthols we proposed, we performed the experiments of the 1-methoxynaphthalene **5** and methyl α-diazoacetate **2** catalyzed by Cu catalysts and obtained a trace amount of the ortho-selective C-H products **6** (Figure 2). Due to the high structural similarity of the reactants, we presuppose that it also follows the same reaction mechanism of naphthols inserted by the Cu carbenes. Since we have discussed the significance of the electrophilic addition and the hydrogen transfer assisted by two water molecules, the DFT calculations are primarily performed to investigate the two crucial elementary steps.

Figure 7 shows the calculated energy profiles of the ortho- and para-C-H bonds of 1-methoxynaphthalenes inserted by the bimetallic and monometallic Cu carbenes. The calculated reaction pathways of the C-H insertion of 1-methoxynaphthalenes are similar to that of naphthols. With the catalysts CuCl or CuCl_2_, the calculated free energy barriers of the electrophilic addition at the ortho-positions of 1-methoxynaphthalenes are higher than the counterparts at the para-positions. It has been emphasized that the H-bonds formed by the hydroxyl groups of naphthols and the bimetallic carbenes are the key factors to determine the site-selectivity in the addition step. Nevertheless, such a specific O-H···Cl H-bond interaction does not exist in the transition states **3-TS-o1** and **4-TS-o1** due to the steric effect of methoxy groups of 1-methoxynaphthalenes.

The energy barriers of the subsequent rate-determining steps, namely the proton transfer steps at the ortho-positions, are relatively higher than that of the para-positions. Specifically, the crucial barriers of the (1,3)-H transfer in the transition states **3-TS-o6-2w** and **3-TS-p6-2w** at the ortho-C-H and para-C-H insertion catalyzed by the (CuCl)_2_ dimer reach high energies of 25.4 and 23.3 kcal mol^−1^, respectively. In another case, the (1,3)-H transfer of the ortho-C-H and para-C-H insertion catalyzed by the CuCl_2_ monomer also have high activation barriers of 24.0 and 22.7 kcal mol^−1^, respectively. The high barriers of both **3-TS-o6-2w** and **4-TS-o6-2w** mean that it is difficult to pass through them and obtain the product at the room temperature when using the trace amount of the product **6** obtained in our experiments. The optimized structures of the transition states **3/4-TS-o6-2w** and **3/4-TS-p6-2w** are remarkably different from that of naphthols due to the lack of the crucial H-bonds formed between the chemical groups of reactants and the metal catalysts.

## 3. Materials and Methods

### 3.1. General Imformation 

^1^H NMR spectra were recorded on a BRUKER 500 spectrometer (Billerica, MA, USA) in CDCl_3_. Chemical reagents were purchased from Leyan (Shanghai, China). Anhydrous dichloromethane (DCM) was distilled from calcium hydride to use. Catalysts CuCl and CuCl_2_ were purchased from Alfa-Aesar Company (Haverhill, MA, USA) and used directly. 

### 3.2. Synthetic Procedure for the Reaction of 1-Methoxynaphthalene and Diazoester

In a dried glass tube, copper catalyst CuCl (0.02 mmol, 5 mol%), 1-methoxynaphthalene **5** (189.6 mg, 1.2 mmol, 3 equiv), and DCM (1 mL) was added at room temperature. Then a solution of methyl phenyl diazaester **2** (76.2 mg, 0.4 mmol) dissolved in 1 mL DCM was introduced into the reaction mixture by a syringe. The resulting mixture was continually stirred at room temperature until product **6** was consumed completely, determined by TLC analysis. After being filtrated through celite and concentrated, the residue was purified by column chromatography on silica gel to obtain the desired product. The yield was determined by ^1^H NMR of crude product by using CH_2_Br_2_ as internal standard.

### 3.3. Computational Methods

All DFT calculations are performed using the Gaussian09 program package [68]. The geometric structures of intermediates and transition states are directly optimized in the solution phase by using the ωB97XD functional [69,70]. The SDD basis set [71] combined with the effective core potential is used to describe the metal element Cu, and the large 6-31 + G** basis set [72] is utilized to describe the nonmetallic elements C, H, O, N and Cl. Frequency analyses are also performed at the same computational level to confirm that the intermediates are local minima and the transition states have only one imaginary frequency. The intrinsic reaction coordinate (IRC) [73,74] calculations are performed to make sure that all transition state structures connect the correct reactants and products in the forward and backward reaction directions. The solvent effect of dichloromethane is evaluated using the SMD [75] model with a dielectric constant ε = 8.93 in Gaussian09. All the calculated energies refer to the Gibbs free energies in the units of kcal mol^−1^ at the temperature of 298.15 K. Further structure details about the intermediates and transition states are provided in Appendix A.

## 4. Conclusions

The detailed mechanisms of the C(sp^2^)-H bond functionalization of naphthols and *α*-aryl-*α*-diazoacetates by the catalysts CuCl and CuCl_2_ are studied through the combined experimental and computational methods. The DFT calculations reveal that the *ortho*-selective products catalyzed by CuCl are obtained from the C-H insertion of naphthols by the bimetallic carbenes. Also, the optimized TS structures of the steps of addition and (1,3)-H transfer reveal that the H-bonds formed by the OH groups of naphthols and the Cl atoms of metal catalysts play an important role in stabilizing the TSs and lowering their energies. In the reaction catalyzed by CuCl_2_, the DFT results indicate that the monometallic carbenes insert into the C(sp^2^)-H bonds of naphthols, rather than the bimetallic species. It is proposed that the H-bond interactions between the Cu carbenes and substrates play an essential role in stabilizing the site-selectivity-determining TSs in all cases, resulting in a lower energy barrier and generating the experimentally observed *ortho*-selective products. The proposed H-bonds assisted insertion by Cu catalysts are supported by our further experiments of the C-H insertion of 1-methoxynaphthalenes catalyzed by the CuCl/CuCl_2_ catalysts as well as the corresponding DFT calculations. Our studies systematically provide the mechanistic insights into the unprecedented C-H functionalization by the CuCl/CuCl_2_ catalysts, which is instructive in designing Cu-catalyzed chemo- and site-selective transformations.

## Data Availability

The data are available on request from the corresponding authors.

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
