# Peer review of "Theoretical Study on the Copper-Catalyzed ortho-Selective C-H Functionalization of Naphthols with α-Phenyl-α-Diazoesters"

_molecules, 2023, doi:10.3390/molecules28041767_

Round 1
Reviewer 1 Report
The authors Zhu et al. report in a combined experimental and theoretical study employing density functional theory results on the functionalization of CH-groups in naphthols and an aromatic ester. The reactions are studied in the liquid phase (dichloromethane) and homogeneously catalyzed by Cu(I) and Cu(II). The authors have carried out a series of mechanistic studies including the calculation of Gibbs free energy barriers (at room temperature) using the Gaussian package. They conclude on the formation of H-bonds which trigger selectivity.
The manuscript is well organized, the language is excellent and the quality of figures and graphs is very high. However, from a technical point of view couple of questions arise, which need to be answered prior to publication of the manuscript.
1) First of all, it is questionable how reliable in terms of accuracy the wB97XD functional is when interactions with Cu(I) and Cu(II) comes into play. The latter is known to be a difficult case and requires careful consideration of electronic correlation effects. The authors need to comment on that. Best would be to compare with a more accurate method.
2) I appreciate the details given in the supporting information, however, one must clearly contradict the statement of the authors that the 6-31G basis set is a large basis set. Especially with respect to the usage of hybrid functionals involving a certain fraction of orbital-dependent Fock exchange, explicit convergence tests using larger basis sets (for instance Ahlrich’s def2 double and triple zeta variants) are mandatory.
3) As the formation of H-bonds is so important for the present research, how do the numerical results depend on the approximation used to describe dispersion interactions. They are expected to contribute in the formation of H-bonds.
These points need to be satisfactorily addressed prior to publication.
Reviewer 2 Report
The article is devoted mostly to the quantum-chemical description of the mechanism of ortho-substitution of naphthols by α-phenyl-α-diazoesters catalyzed by CuCl and CuCl2. Under standard conditions, the preferred way in the such reaction is para-substitution. However, it was shown experimentally earlier that the use of CuCl and CuCl2 helps to achieve ortho-selectivity. The authors considered the reaction paths from reactants to products throughout the transition states for both these ways and showed with help of calculated thermodynamics parameters why ortho-substitution is preferable when copper chlorides are used as catalysts.
I believe the presented work is very interesting and useful for the field of purposeful synthesis of new chemicals, therefore it can be considered for publication. However, as a minor correction, the reasons for the quantum-chemical method (DFT functional and basis sets) choice should be described. It is usual practice to demonstrate that chosen method is suitable for the considered compounds and their properties description.
Reviewer 3 Report
The manuscript is devoted to a theoretical study of a very complex catalytic reaction involving copper salts. Previous experimental work has led to the development of a convincing reaction, justified by adequate quantum chemical methods.
At the same time, in the abstract, in the introduction, and in the conclusion sections of this manuscript, it is said that significant results were obtained based on experimental and theoretical studies, although the experimental part does not contain a description of the experiments.
In this regard, the authors should clarify the purpose of their theoretical work, based on their previous experimental data, confirming the previously stated hypotheses, on the basis of which large-scale generalizing conclusions about the mechanism of the studied reaction were made.
The article is recommended for publication after minor revision.
Round 2
Reviewer 1 Report
After having read the revised version of the manuscript, I conclude that the authors amended the manuscript sufficiently. Based on the revisions I can recommend publication in its present form.